# Analysis of the specificity of a COVID-19 antigen test in the Slovak mass testing program

**Michal Hledík[1]☯, Jitka Polechová[2]☯, Mathias Beiglböck[2], Anna Nele Herdina[3], Robert Strassl[3], Martin Posch[4]***

**1** Institute of Science and Technology Austria (IST Austria), Klosterneuburg, Austria, **2** Department of Mathematics, University of Vienna, Vienna, Austria, **3** Division of Clinical Virology, Department of Laboratory Medicine, Medical University of Vienna, Vienna, Austria, **4** Center for Medical Statistics, Informatics, and Intelligent Systems, Medical University of Vienna, Vienna, Austria

☯ These authors contributed equally to this work.
* martin.posch@meduniwien.ac.at

**Data Availability Statement:** The data underlying the results presented in the study are available from Institute for Healthcare Analyses (IZA) of the Ministry of Health of the Slovak Republic on GitHub

## Abstract

### Aims

Mass antigen testing programs have been challenged because of an alleged insufficient specificity, leading to a large number of false positives. The objective of this study is to derive a lower bound of the specificity of the SD Biosensor Standard Q Ag-Test in large scale practical use.

### Methods

Based on county data from the nationwide tests for SARS-CoV-2 in Slovakia between 31.10.–1.11. 2020 we calculate a lower confidence bound for the specificity. As positive test results were not systematically verified by PCR tests, we base the lower bound on a worst case assumption, assuming all positives to be false positives.

### Results

3,625,332 persons from 79 counties were tested. The lowest positivity rate was observed in the county of Rožňava where 100 out of 34307 (0.29%) tests were positive. This implies a test specificity of at least 99.6% (97.5% one-sided lower confidence bound, adjusted for multiplicity).

### Conclusion

The obtained lower bound suggests a higher specificity compared to earlier studies in spite of the underlying worst case assumption and the application in a mass testing setting. The actual specificity is expected to exceed 99.6% if the prevalence in the respective regions was non-negligible at the time of testing. To our knowledge, this estimate constitutes the first bound obtained from large scale practical use of an antigen test.

(https://github.com/Institut-Zdravotnych-Analyz/covid19-data).

**Funding:** The author(s) received no specific funding for this work.

**Competing interests:** NO authors have competing interests.

## Introduction

While PCR-tests are usually considered as the gold standard to detect infection with the SARS-CoV-2 coronavirus in terms of sensitivity as well as specificity, antigen tests (Ag-Tests) offer practical advantages in terms of costs, logistics and speed [1]. Because Ag-Tests may play a major role in large scale testing strategies [2, 3] in populations with low prevalence, besides their sensitivity also the specificity is of significant interest, especially as resulting low positive prognostic values may lead to confusion and public distrust into the testing strategy. The objective of this study is to obtain a lower bound of the specificity of the SD Biosensor Standard Q Ag-Test based on data of mass tests in Slovakia and infer Ag-Test specificity from a large sample of the general population. The lower bound for the specificity is obtained by making the conservative assumption that potentially all positive results are false positive results. This is in contrast to other studies which use a PCR-test as a reference to estimate sensitivity and specificity of an Ag-Test [4–7].

## Methods

From late October to early November 2020, Slovakia undertook large scale testing of its population [8]. Participants were tested at specially set up locations by medical personnel via nasopharyngeal swab sampling and received their results after a short waiting period. For testing, the SD Biosensor Standard Q Ag-Test was used [9, 10]. The tests in Slovakia were divided into three phases. In the first pilot phase, testing was only conducted in certain particularly affected counties. In Phase 2 (31.10.–1.11.) all Slovak counties were tested. In Phase 3 (6.11.–8.11.) all heavily affected counties (those with > 0.7% prevalence during Phase 2) were tested again. In this retrospective study, we use publicly available data on the outcome of tests in Phase 2 on county level [11]. Participation in testing was voluntary, but it was a condition to avoid quarantine. Persons that were quarantined due to a previous positive PCR-test for COVID-19 or due to being a close contact of such a person were excluded from the test.

## Statistics

To derive a lower bound for the specificity we made the conservative ("worst case") assumption that potentially all positive results constitute false positive results. For each of the 79 counties, we compute the rate of positive tests together with two-sided binomial Clopper-Pearson confidence intervals at the Bonferroni-adjusted significance level of alpha = 0.05/79 (adjusted for the number of counties; see e.g. [12]). The minimum upper bound of these adjusted confidence intervals is an upper 97.5% confidence bound for the test positivity rate. This will be in a county with low disease activity and a large sampled population. Under the conservative assumption that all positive results constitute false positives, it is also an upper bound on the false positive rate of the test and defines a lower bound for the true specificity. Denoting prevalence, sensitivity and specificity by $r$, $s$ and $p$, this holds under the assumption that $s + p \geq 1$ (i.e., that the probability of a positive test result is larger for a SARS-COV2 positive than for a SARS-COV2 negative subject). This is fulfilled for any test that performs not strictly worse than chance. Then the overall rate of positive results is given by $q = r\,s + (1 - r)(1 - p)$ and it follows that $q \geq 1 - p$, where $1 - p$ is the false positive rate of the test. Consequently, $1 - q$ is a lower bound for the specificity $p$.

## Results

In phase 2, all residents aged 10 to 65 throughout Slovakia were invited to get tested and 3 625 332 participated (about 66% of the population of Slovakia). Among 79 administrative counties

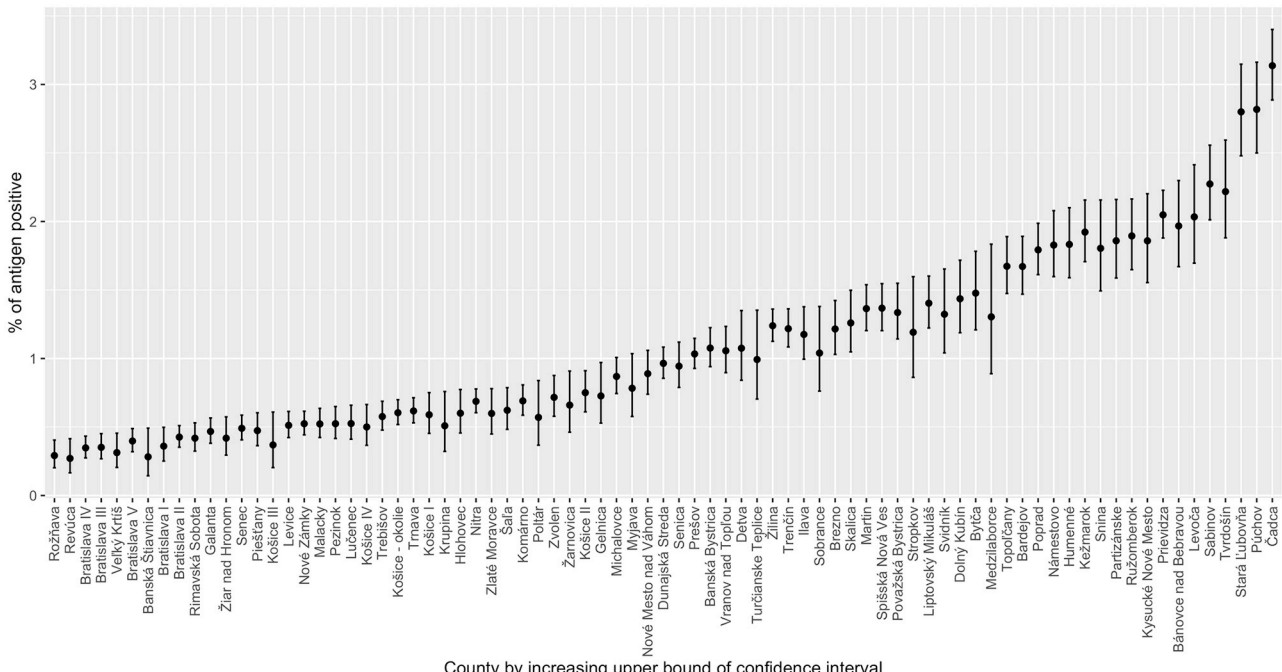

**Fig 1. SARS-CoV-2 antigen detected incidence in Slovakian mass testing by county.** Test positivity rates among the 3 625 332 tested persons in the 79 counties with simultaneous Bonferroni adjusted 95% confidence intervals. Counties are ordered from low to high incidence by the upper confidence bound. Data source: [11]. The corresponding data are tabled in S1 Table.

of Slovakia, participation rate varied from 39% (Košice III county) to 78% (Senec county) of inhabitants. Fig 1 depicts the test positivity rate of the individual counties ordered according to the upper bound of the simultaneous Bonferroni adjusted 95% confidence interval from low to high incidence. The lowest upper bound (obtained for Rožnava) is 0.40% (see Table 1). In terms of specificity (instead of false positives) this implies that with 97.5% confidence, the specificity of Standard Q is higher than 99.6%. As a sensitivity analysis we also consider the five counties with the lowest upper bound for the positivity rate. All imply specificities above 99.54%. These counties include regions with both relatively low participation rate (Rožňava, Revúca and Veľký Krtíš rank 65., 67. and 57. in participation among the 79 counties) and relatively high participation rate (Bratislava IV and Bratislava III rank 13. and 4.).

**Table 1. The five counties with the lowest upper bound on the positivity rate of antigen tests.**

| County | Positivity rate upper bound simultaneous 95% CI | Number of tests | Number of positive tests | Participation, % of inhabitants tested | Population 2019 [13] |
|---|---|---|---|---|---|
| Rožňava | 0.40% | 34307 | 100 | 55% | 62131 |
| Revúca | 0.41% | 21419 | 58 | 54% | 39537 |
| Bratislava IV | 0.43% | 65861 | 229 | 67% | 97792 |
| Bratislava III | 0.45% | 49788 | 175 | 72% | 69479 |
| Veľký Krtíš | 0.46% | 24282 | 76 | 56% | 43263 |

## Discussion

In this study the lower bound for the specificity of the SD Biosensor Standard Q AG test was 96.4%. This worst case analysis provides only a lower bound of the true specificity since it neglects entirely the true incidence of COVID-19. As the antigen tests in Slovakia were not controlled directly using accompanying PCR-tests, we refrain from an attempt to subdivide the observed positives into true and false positives.

The above estimate is consistent with the information provided by the manufacturer [14], stating 0.32% (0.01, 1.78) as a false positive rate—a very broad CI. The derived upper bound from the mass testing is more informative and appears relevant in that it constitutes the first (to the best of our knowledge) bound obtained from large scale practical use of Standard Q and also suggests better performance of Standard Q compared to previous studies: a large study (with 2347 SARS-CoV-2-free samples based on a PCR-test), suggests a specificity of 99.3% (CI 98.6–99.6) [7]. Other available specificity estimates are based on an order of magnitude smaller samples (99.2% (CI 97.1–99.8) [4] and 100% [6] and, in addition, are from study populations with high incidence rates (according to PCR-testing). A point estimate of 98.53% is given by [5], based on 100 SARS-Cov2-free samples with other respiratory viruses present and 35 samples from healthy volunteers.

Data from Phase 2 was used in this study, because the general countrywide testing was performed irrespective of regional incidence rates, covering also regions with potentially very low incidence. The data has also been used to investigate spatial patterns of the spread of COVID-19 [10] and to demonstrate that both mass-scale testing and restrictive measures contributed to the sharp drop in COVID-19 incidence [9]. However, to avoid a positively biased specificity estimate, compared to [9] we use an updated, more detailed dataset, where an erroneous positivity rate for Bratislava IV has been corrected. The binomial distribution assumption underlying the computation of the Clopper-Pearson confidence intervals, is based on the assumption of independent events that could be violated if, e.g. testing stations were operating with different quality. However, the Bonferroni-correction is a strictly conservative approach to derive simultaneous confidence bounds that account for the selection of the county with the lowest upper bound to derive the estimate. It is well understood in the epidemiology literature that imperfect reference criteria lead to a systematic bias of the true incidence [15]. Specifically, the sensitivity and specificity of tests are systematically underestimated if the criterion (SARS-CoV-2 infection) cannot be assessed directly and tests are compared to a surrogate criterion as gold standard which is subject to errors. This could be a contributing factor for the relatively higher specificity found in Slovak data compared to other studies. However, there are other conceivable factors that could bias the results: different quality of swab sampling and handling, low temperature in outdoor testing stations, deviations in the production over time and data quality.

While the analysis is based on a worst case assumption, considering all positives to be false positives, the obtained estimate appears relevant in that it constitutes the first bound obtained from large scale practical use of Standard Q. It suggests better specificity of Standard Q compared to previous studies and these findings can support the planning and justification of future mass testing programs. Besides test specificity, the sensitivity is essential for the evaluation of the overall utility of mass testing strategies. Unfortunately, based on the mass testing data set no estimate of the test sensitivity can be obtained and, to our knowledge, no sensitivity estimates have been provided in a mass-testing setting so far such that further studies are needed.

## Supporting information

**S1 Table. County, number of tests, number of positive tests, positivity rates, simultaneous Bonferroni adjusted 95% confidence intervals, and standard errors of the positivity rates in the 79 counties.** Counties are ordered from low to high incidence by the upper confidence bound. Data source [11].
(PDF)

## Acknowledgments

We would like to thank Alfred Uhl, Richard Kollár and Katarína Bod'ová for very helpful comments. We also thank Matej Mišík for discussion and information regarding the Slovak testing data and Ag-Test used.

## Author Contributions

**Conceptualization:** Jitka Polechová, Mathias Beiglböck.

**Data curation:** Michal Hledík, Jitka Polechová.

**Formal analysis:** Michal Hledík, Martin Posch.

**Investigation:** Michal Hledík, Jitka Polechová.

**Methodology:** Michal Hledík, Jitka Polechová, Mathias Beiglböck, Anna Nele Herdina, Robert Strassl, Martin Posch.

**Software:** Michal Hledík, Martin Posch.

**Supervision:** Mathias Beiglböck.

**Validation:** Michal Hledík, Jitka Polechová, Martin Posch.

**Visualization:** Michal Hledík, Martin Posch.

**Writing – original draft:** Michal Hledík, Jitka Polechová, Mathias Beiglböck, Anna Nele Herdina, Robert Strassl, Martin Posch.

**Writing – review & editing:** Michal Hledík, Jitka Polechová, Mathias Beiglböck, Anna Nele Herdina, Robert Strassl, Martin Posch.

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
