## [Decision Letter · Decision Letter 0]

26 May 2021

PONE-D-21-14542

Analysis of the specificity of a COVID-19 antigen test in the Slovak mass testing program

PLOS ONE

Dear Dr. Posch,

Thank you for submitting your manuscript to PLOS ONE. After careful consideration, we feel that it has merit but does not fully meet PLOS ONE’s publication criteria as it currently stands. Therefore, we invite you to submit a revised version of the manuscript that addresses the points raised during the review process.

I obtained the comment from one reviewer but could not obtain it from the other reviewer. To progress the smooth review process, I have determined to move to the next round for the review.

We look forward to receiving your revised manuscript.

Kind regards,

Etsuro Ito

Academic Editor

PLOS ONE

Journal Requirements:

3. We note you have included a table to which you do not refer in the text of your manuscript. Please ensure that you refer to Table 1 in your text; if accepted, production will need this reference to link the reader to the Table.

Reviewers' comments:

Reviewer's Responses to Questions

**Comments to the Author**

1. Is the manuscript technically sound, and do the data support the conclusions?

Reviewer #1: Yes

2. Has the statistical analysis been performed appropriately and rigorously? 

Reviewer #1: Yes

3. Have the authors made all data underlying the findings in their manuscript fully available?

Reviewer #1: Yes

4. Is the manuscript presented in an intelligible fashion and written in standard English?

Reviewer #1: Yes

5. Review Comments to the Author

Reviewer #1: In presenting argument in the introduction that antibody testing can be an alternative for RT-CPR, it may be helpful to note that the test evaluates different things: active infection vs history of infection (including current), with different delays between tests being positive (longer for serology). This reviewer agrees that serology can be a very attractive alternative for reasons cited by the authors, but additional caveats other than SN (sensitivity) and SP (specificity) related to meaning of the test in terms of inference of onset of incident infection should be clearly presented.

Can you please be clearer in the introduction how your approach is “in contrast to other studies which use a PCR-test as a reference to estimate sensitivity and specificity of an Ag-Test4”?

One alternative to Bonferroni correction that seems appealing to this reviewer is to model rate of positive tests using binomial regression with random effect of county, and maybe some fixed effects that can account for rate of positive tests, such as numbers quarantined per county, positivity rates from Phase 1, age structure, economic indicators, percent of county tested, outdoor temperature, count/proportion of outdoor test sites, date of test (the usual confounders in epidemiology that can affect both willingness to test and chance of having been infected plus those mentioned in the discussion by the authors). The fixed intercept of the model would overall rate of positive tests and can be made county-specific through use of other random and fixed covariates; upper percentile of these modelled estimates can be used authors did with their Bonferroni-adjusted confidence interval. One may be also tempted to model spatial correlation, if descriptive statistics support its existence.

Instead of just giving upper bound of 95%CI, it would be more informative to give both the estimate per county and associated standard error. This should allow anyone to estimate different percentile if for their purposes 97.5% is not appropriate. This may already be in the supplemental materials, so maybe this is just a matter of pointing the reader in the right direction.

6. PLOS authors have the option to publish the peer review history of their article (what does this mean?). If published, this will include your full peer review and any attached files.

Reviewer #1: **Yes: **Igor Burstyn

---

## [Author Response · Author response to Decision Letter 0]

12 Jul 2021

We thank the referee for the helpful remarks and revised the manuscript accordingly. In addition, we added a table with all estimates in the supporting information and formatted the document according to the PLOS ONE formatting guidelines. Please see the submitted point to point answer for details.

---

## [Editor Report · Decision Letter 1]

14 Jul 2021

Analysis of the specificity of a COVID-19 antigen test in the Slovak mass testing program

PONE-D-21-14542R1

Dear Dr. Posch,

We’re pleased to inform you that your manuscript has been judged scientifically suitable for publication and will be formally accepted for publication once it meets all outstanding technical requirements.

Kind regards,

Etsuro Ito

Academic Editor

PLOS ONE

---

## [Editor Report · Acceptance letter]

21 Jul 2021

PONE-D-21-14542R1 

Analysis of the specificity of a COVID-19 antigen test in the Slovak mass testing program 

Dear Dr. Posch:

I'm pleased to inform you that your manuscript has been deemed suitable for publication in PLOS ONE. Congratulations! Your manuscript is now with our production department. 

Kind regards, 

on behalf of

Prof. Etsuro Ito 

Academic Editor

PLOS ONE